# Mitigating the Drying Shrinkage and Autogenous Shrinkage of Alkali-Activated Slag by NaAlO_2_

**DOI:** 10.3390/ma13163499

**Published:** 2020-08-08

**Authors:** Bin Chen, Jun Wang, Jinyou Zhao

**Affiliations:** College of Civil Engineering, Northeast Forestry University, Harbin 150040, China; chenbin@nefu.edu.cn

**Keywords:** alkali-activated slag, drying shrinkage, autogenous shrinkage, mitigation

## Abstract

The shrinkage of alkali-activated slag (AAS) is obviously higher than ordinary Portland cement, which limited its application in engineering. In this study, the effects of NaAlO_2_ in mitigating drying shrinkage and autogenous shrinkage of AAS were studied. To further understand the shrinkage mechanism, the hydration products and microstructures were studied by X-ray diffraction, scanning electron microscopy and nitrogen adsorption approaches. As the partial substitution rate of NaAlO_2_ for Na_2_SiO_3_ increased, the drying shrinkage and autogenous shrinkage reduced significantly. The addition of NaAlO_2_ could slow down the rate of hydration reaction and reduce the porosity, change the pore diameter and the composition of generated paste and cause more hydrotalcite and tetranatrolite generated—which contributed to reduced shrinkage. Additionally, raising the Na_2_O content rate caused obvious differences in drying shrinkage and autogenous shrinkage. As the Na_2_O content elevated, the drying shrinkage decreased and autogenous shrinkage increased. A high Na_2_O content would cause complete hydration reactions and provoke high autogenous shrinkage. However, incomplete hydration reactions left more water in the paste, and the evaporated water dramatically influenced drying shrinkage. The results indicate that addition of NaAlO_2_ could greatly mitigate the drying shrinkage and autogenous shrinkage of AAS.

## 1. Introduction

Alkali-activated slag (AAS)-replaced ordinary Portland cement as a new binding material could be considered as possibility [1]. Realized by compounded blast furnace slag and alkaline activator was obtained the binding materials which is similar to cement. AAS could effectively reduce carbon dioxide emissions and energy consumption, while also effectively used industrial solid waste [2,3]. The AAS had good performance in early compressive strength, high temperature resistance and chemical corrosion resistance [4,5]. However, unsolved problems were existed, such as higher drying shrinkage and autogenous shrinkage [6,7].

Previous research has demonstrated that the shrinkage rate of AAS was significantly higher than that of ordinary Portland cement. The study by Lee et al. [8] provided that the AAS autogenous shrinkage and drying shrinkage at 28 d was about four and six times to that of ordinary Portland cement, respectively. Atiş et al. [9] reported that the AAS mortar drying shrinkage prepared with Na_2_SiO_3_ as an alkaline activator was three to six times to that of ordinary Portland cement mortar. Drying shrinkage of cementitious materials has already been confirmed that mainly due to the capillary stress, disjoining pressure and surface tension. Cartwright et al. [10] believed that the capillary stress was caused by the smaller pore diameter, which led to the AAS shrinkage. In addition, the C–A–S–H in the AAS hydration product was considered to be the main reason for the large shrinkage. The alkali metal cations in the reaction solution entered the C–A–S–H gel, changed structures and led to collapse and redistributed upon drying [11,12].

High drying shrinkage and high autogenous shrinkage deformation could seriously influence the properties and durability of AAS, which limited the usage and promotion of AAS. Therefore, how to effectively mitigate the drying shrinkage and autogenous shrinkage of AAS became an important issue that must be solved urgently. Previous studies have found that the pore structure could mitigate shrinkage by adding different admixtures [13,14,15]. The main solutions to effectively mitigate the shrinkage of AAS were to reduce porosity, slow down hydration reaction rates or increase the stiffness of the microstructure. Abdel-Gawwad [16] demonstrated that the addition of Al_2_O_3_ in AAS produced a hydrotalcite structure with micro-swelling properties, which could effectively mitigate shrinkage. By changing the dose of Al^3+^ in the reaction solution, he could influence the hydration reaction rate. Haha [17] and Sakulich et al. [18] suggested that the addition of Al_2_O_3_ slowed down the hydration reaction rate—and the formation of hydrotalcite structure could effectively mitigate AAS shrinkage. According to extensive reports, the amount of Al^3+^ in the reaction solution can change the hydration reaction rate, while also significantly changing the microstructure. Therefore, this research proposes to use NaAlO_2_ partially substituted Na_2_SiO_3_ as a composite alkaline activator. The partial substitution of NaAlO_2_ for Na_2_SiO_3_ could provide Al^3+^ to the reaction solution; Al^3+^ was chosen as it has the ability to delay the hydration reaction rate and change the microstructure, which are beneficial for mitigating the AAS shrinkage. At the same time, NaAlO_2_ could provide the OH^-^, which was the required by the hydration reaction power and crucial for the AAS properties development. Moreover, considering the cumbersome procedures during the preparation of the liquid alkaline activator, it was proposed to use solid NaAlO_2_ and Na_2_SiO_3_ as composite solid alkaline activator, which would effectively improve efficiency and alleviate the alkali corrosion during the preparation and curing process. In short, the study of NaAlO_2_ partially substituted Na_2_SiO_3_ as a composite alkaline activator to prepare AAS could be a good idea to mitigate drying shrinkage and self-shrinkage. The study of hydration mechanism of mitigated shrinkage would be extremely important for the preparation of excellent AAS.

This study was aimed to study the effects of mitigating shrinkage by NaAlO_2_. In this study, we studied the effects of four different NaAlO_2_ substitution rates (for 20%, 40%, 60% and 80%) and five different Na_2_O contents (for 3%, 5%, 7%, 9% and 11%, which substitution rate fixed 20%) of drying shrinkage and autogenous shrinkage. Furthermore, the hydration products, microstructures and pore structure were analyzed by X-ray diffraction (XRD), scanning electron microscopy (SEM) and nitrogen adsorption approach (Brunauer–Emmett–Teller theory), in order to understand the mechanism of hydration of mitigated shrinkage.

## 2. Materials and Methods

### 2.1. Raw Material and Mixture Ratio

Blast furnace slag with specific surface of 435 m^2^/kg was obtained from Minmetals Yingkou Medium Plate (Yingkou, China). According to the GB/T 18046-2017 [19], the slag was classified as S95. The chemical components are shown in Table 1. NaAlO_2_ (analytical pure) was provided by Dalu Chemical Reagent Factory (Tianjin, China). Na_2_SiO_3_ (Na_2_SiO_3_·9H_2_O, ratio of Na_2_O to SiO_2_ content was 1.03 ± 0.03) was provided by Xilong Scientific Chemical Reagent Factory (Shantou, China).

Four different NaAlO_2_ substitution rates (for 20%, 40%, 60% and 80%), five different Na_2_O contents (for 3%, 5%, 7%, 9% and 11%, which substitution rate fixed 20%) and a non-substituted control group were tested. The prepared mix proportions of AAS are shown in Table 2. For example, the code “NA0.2NS0.8-5” represents the alkaline activator with NaAlO_2_ substitution rates of 20% and Na_2_SiO_3_ content of 80% and Na_2_O contents 5%. In order to reduce the influence of dosage to the test error, the slag was fixed to 2000 g and the water–slag ratio was kept constant at 0.35.

### 2.2. Test Methods

#### 2.2.1. Drying Shrinkage

The drying shrinkage test was on the basis of JC/T 603-2004 [20]. All prepared paste were poured into prismatic molds (h = 25 mm, w = 25 mm, l = 280 mm). The sample was cured and measured in the curing room (20 ± 2 °C and RH > 50%). The test start time was recorded from demolding done and the measurement accuracy was 0.001 mm. The drying shrinkage was calculated according to Equation (1). Three samples were prepared for each code and the drying shrinkage was the mean value of three samples and recorded once per day until 28 d.
(1)$$S_{n}=\frac{\left(L_{0}-L_{n}\right)\times 100}{280},$$
*S_n_* was the drying shrinkage rate of the test at the age of *n* d and the unit was %. *L*_0_ was the initial reading and the unit was mm. *L_n_* was the measurement of the test at the age of *n* d and the unit was mm. The effective length of the fixed value and the unit was 280 mm.

#### 2.2.2. Autogenous Shrinkage

An autogenous shrinkage test was under the standard ASTM C1697-09 [21]. This solution could effectively avoid moisture loss and minimize the constraints on volume changes. All prepared pastes were poured into a corrugated tube of 420 mm (Ø28.5 mm). The loaded samples were placed horizontally on a corrugated plastic board to avoid length changes and damage. The samples were cured and measured in the curing room (20 ± 2 °C and RH > 50%). The start time of the test was recorded from the time of adding water to mix. The measurement accuracy was 0.001 mm. The autogenous shrinkage was calculated according to Equation (2). Three samples were prepared for each code, and the drying shrinkage was the mean value of three samples and recorded once per day until 28 d.
(2)$$\mu_{n}=\frac{\left(L_{n}-L_{0}\right)\times 10^{6}}{L_{0}},$$
*µ_n_* was the autogenous shrinkage rate of the test at the age of *n* d, and the unit was µε. *L_n_* was the measurement of the test at the age of *n* d, and the unit was mm. *L*_0_ was the initial reading and the unit was mm.

#### 2.2.3. Instrumental Techniques

In order to better understand the shrinkage performance of AAS with different mix ratios, scanning electron microscopy (SEM, JEOL-6360LV, Tokyo, Japan), X-ray diffraction (XRD, Empyrean-X, Amsterdam, The Netherlands) and Brunauer–Emmett–Teller (BET, Micromeritics ASAP 2020 instrument, Pullman, WA, USA) were used to analyze microstructures, hydration products and pore structures. The specimens from the crushed samples were poured into alcohol for 24 h to discontinue hydration. The specimens were removed from the alcohol and dried in a 60-°C oven for use before testing.

## 3. Results and Discussion

### 3.1. Drying Shrinkage

Figure 1 plots the effects of different NaAlO_2_ substitution rates on drying shrinkage. Drying shrinkage mainly occurred in the first 7 d and tended to increase steadily and slowly after 7 d. At 7 d, the drying shrinkage of the NaAlO_2_ substitution rate of 20%, 40%, 60% and 80% was 93%, 84%, 64% and 54% of control group (NS 1.0-5), respectively. The substitution of NaAlO_2_ for Na_2_SiO_3_ had a significant effect on mitigate shrinkage. Substitution rate of NaAlO_2_ has a significantly negative correlation with drying shrinkage. The main reason was that the rate of drying shrinkage was clearly related to the hydration reaction degree of the paste [22]. Na_2_SiO_3_ had stronger alkali activating ability. Neto et al. [23] noted that more Na_2_SiO_3_ content in the alkaline activator caused clear high drying shrinkage. Meanwhile, Ben et al. [17] reported that more soluble Al in the paste could delay the alkali-activated hydration reaction, which could contribute to mitigate drying shrinkage.

Figure 2 shows the effects of different Na_2_O contents on drying shrinkage at fixed 20% NaAlO_2_ substitution rate. The drying shrinkage was also obvious during the first 7 d. At 7 d, Na_2_O content of 5%, 7%, 9% and 11%, the drying shrinkage was 83%, 73%, 56% and 30% of Na_2_O content of 3%, respectively. The Na_2_O content had a clear effect on drying shrinkage: a high Na_2_O content led to low drying shrinkage. As result of low Na_2_O content with low OH^-^ ions concentration, the paste could not complete hydration reaction and residual moisture evaporation led to high drying shrinkage [24]. Meng Wu et al. [25] also suggested that amorphous gels lost moisture caused greater drying shrinkage. In general, the mechanism of AAS drying shrinkage was complicated and depended on different NaAlO_2_ substitution rates and Na_2_O contents.

### 3.2. Autogenous Shrinkage

Figure 3 shows the effects of different NaAlO_2_ substitution rates on autogenous shrinkage. Used bellows to test autogenous shrinkage could effectively prevent the loss of water and minimize the constraints on volume changes. Autogenous shrinkage had similar characteristics to drying shrinkage and great influence to mitigate autogenous shrinkage by the NaAlO_2_ substitution rate increased and also was obvious shrank during the first 7 d. Moreover, the autogenous shrinkage of alkali-activated slag used alkaline activator with Na_2_SiO_3_ alone was still very large, same to findings from by Kumarappa et al. [26]. The sample with substitution rate at 80%, the drying shrinkage at 7 d and 28 d were 73% and 72% of the control group (NS 1.0-5), respectively.

Figure 4 plots the effects of different Na_2_O contents on autogenous shrinkage at fixed 20% NaAlO_2_ substitution rates. By elevating the Na_2_O content, the autogenous shrinkage increased, consistent with the Ma et al. [27] found. However, the results were contrary to the drying shrinkage test results. The autogenous shrinkage happened without the exchange of external substances and the water from paste was not migrated. Hence, high Na_2_O content promoted completed the hydration reaction of the paste, which in turn caused greater autogenous shrinkage [28]. The maximum autogenous shrinkage group (NA0.2NS0.8-11) was 1.4 times and 1.5 times that of the minimum autogenous shrinkage group (NA0.2NS0.8-3) at 7 d and 28 d, respectively. Then, the phenomenon of autogenous shrinkage with different Na_2_O contents would be consistent with the degree of hydration reaction. Neto et al. [23] also believed that drying shrinkage due to the water evaporation from the paste was more significant than autogenous shrinking.

### 3.3. XRD

The composition of the hydration products was analyzed by XRD. The generation of different hydration products had significant impacts on the pore structure and shrinkage properties. By checking the results of drying shrinkage and autogenous shrinkage, we found that drying shrinkage and autogenous shrinkage before 7 d were more severe than that of after 7 d. Therefore, the characteristics of hydration products at 7 d could explain the different shrinkage changes. Figure 5 shows the XRD image of raw slag and different NaAlO_2_ substitution rates at 7 d. The characteristic phenomenon at approximately 30° of all AAS was a feature corresponding to big quantity of amorphous products which were C-(A)-S–H gels [29]. Calcite was found in all pastes and was generated by sample carbonization during preparation and curing time [30]. However, hydrotalcite and tetranatrolite were also major products found in this research. Hydrotalcite had micro-expansion in a certain degree; tetranatrolite was a zeolite phase structure, which was beneficial to mitigate paste shrinkage [31]. At the same time, as the substitution rate of NaAlO_2_ increased, the amount of hydrotalcite and tetranatrolite also increased, which generated different shrinkage mitigation effects among all the AAS—and was consistent with the results of the drying shrinkage test.

### 3.4. SEM

The microscopic structure of AAS with different NaAlO_2_ substitution rates at 7 d and 28 d were analyzed by SEM. The progress of hydration reaction was investigated by studying the microscopic morphology. The density of hydration products and the changes of voids could be retrieved directly from the SEM, which contributed to better understanding of the mechanism of hydration. Figure 6 shows the microscopic structure of AAS with the different NaAlO_2_ substitution rates at 7 d and 28 d. The specimen NS1.0-5 was significantly denser than that of NA0.2NS0.8-5 and NA0.8NS0.2-5. The slag was almost completely dissolved and generated more C–A–S–H gels and less voids at 7 d. The Na_2_SiO_3_ was conducive to the progress of the hydration reaction and could make the hydration reaction more complete. This showed that NaAlO_2_ partially substituted Na_2_SiO_3_ as a composite alkaline activator, inhibiting the progress of hydration reaction, delaying and reducing the formation of hydration products of AAS, which supports the conclusions of Ben et al. [31].

The phenomenon of microstructure at 28 d was exactly opposite of that at 7 d. The specimen NA0.8NS0.2-5 generated denser structure (for mainly C–A–S–H gels) at 28 d, and the microstructure was denser than that of NS1.0-5. This indicates that NA0.8NS0.2-5 had a more complete hydration reaction at 28 d. Moreover, the NaAlO_2_ substitution rates difference had significant impact on the formation of 28 d hydration products. This may be due to the addition of NaAlO_2_ having an inhibitory effect on the hydration reaction in the early phase, but a motivating effect on the reaction in the later phase, which supports the research of Huang et al. [32]. The NA0.8NS0.2–5 clearly had completed hydration reaction to generate more gels where micropores mainly existed. Compared with other samples, NA0.8NS0.2-5 had the most amount of gels, which meant a higher micropore-to-volume ratio was excited, which corresponded to the BET results.

### 3.5. BET

Many studies found that the volume proportions of pore structure have critical impact on the shrinkage properties of AAS [33,34]. The pore size was determined to measure by the Brunauer–Emmett–Teller (BET) theory. The specimens at 28 d were tested to observe the volume proportion of AAS pore structure. The pore sizes were in the range of 1.7–300 nm. According to Zhongwei Wu, a Chinese academic, the pore structure of cement-based materials be classified as micropores (d < 20 nm), mesopores (20 nm < d < 50 nm) and macropores (d > 50 nm). The volume proportions of pore structure distribution of AAS showed in Table 3.

The pore-size distributions of AAS with different NaAlO_2_ substitution rates are shown in Figure 7, mainly concentrated between 1.7–20 nm. Based on Figure 7, a higher NaAlO_2_ substitution rate resulted to lower total proportion of porosity and higher micropores volume. The reduction of total porosity was beneficial to mitigate drying shrinkage. When the NaAlO_2_ substitution rates were 20% and 80%, the total porosities were 56% and 51% of the control group (NS1.0-5), respectively, which was consistent with the drying shrinkage and SEM results. On the other hand, the adjustment in pore-size distribution also had significant effects on the drying shrinkage. The substitution rate increased with higher micropores volume proportion, which was the reason the drying shrinkage reduced. The highest substitution rate corresponded to the largest micropores volume proportion of 61.67%, which also corresponded with the smallest drying shrinkage. This indicates that the micropore volume proportion had a significant effect on drying shrinkage. Compare this to the discussion by Chen et al. [35], namely, that that pore structure also has significant effect on drying shrinkage: a higher micropore volume proportion can result in a greater drying shrinkage, which was contrary to the phenomenon found in this study. The reason for the difference in Chen’s conclusion was that the hydration products had no changes. However, in this study, the addition of NaAlO_2_ changed the hydration products. Based on the findings from the XRD results, more hydrotalcite and tetranatrolite were generated. Nevertheless, the micropores only existed in the gels, so the micropores volume proportion increased may correspond to the increased in the amount of hydrotalcite and tetranatrolite [36]. As discussed in this article, both hydrotalcite and tetranatrolite contributed to mitigate drying shrinkage and generated lower drying shrinkage with more micropores volume proportion.

The critical pore size was determined by examining the differential pore volume curves. The pore-size distribution of the code of NS1.0-5 and NA0.2NS0.8-5 were multimodal pore distributions, but the code of NA0.8NS0.2-5 was observed to have a unimodal pore distribution. The critical pore diameter of control group (NS1.0-5) was 46 nm; the critical pore diameter of NA0.2NS0.8-5 and NA0.8NS0.2-5 was reduced to 20 nm and 16 nm, respectively. As the substitution rate of NaAlO_2_ increased, the value of the critical pore diameter decreased, which led to less shrinkage. Kumarappa [26] had a same conclusion. In addition, the order of critical pore size had a similar trend to the total porosity (NS1.0-5 > NA0.2NS0.8-5 > NA0.8NS0.2-5).

The pore-size distributions of AAS with different Na_2_O contents are shown in Figure 8. As Na_2_O content increased, the total porosity and micropores volume proportion decreased. The content of Na_2_O content increased from 3 to 11%. The micropore volume proportions decreased from 75.92% to 16.24%. A higher Na_2_O content led to more complete reaction, and the paste became denser and the total porosity decreased, which supports the research of Chindaprasirt et al. [37]. At this point, the largest micropores volume proportion corresponded to the maximum drying shrinkage—the same as concluded by Yang et al. [38]. The micropores of diameter 1.25–25 nm were the main reason for AAS drying shrinkage. However, contrary to the conclusions obtained by the different substitution rates above, the major reason was that the NaAlO_2_ substitution rate was fixed and changed the Na_2_O content could not change the composition of the generated phase. More micropore volume proportions in the same generated paste led to greater drying shrinkage.

Furthermore, as the Na_2_O content elevated, the critical pore size became smaller, which corresponded to the lower shrinkage and was consistent with the results of the different NaAlO_2_ substitution rates.

## 4. Conclusions

In this study, the mitigation function of alkali-activated slag-drying shrinkage and autogenous shrinkage mixed with NaAlO_2_ was studied. The mechanism and reason of mitigating shrinkage were discussed. According to the research in this article, the conclusions are as follows:Through NaAlO_2_ addition, the drying shrinkage and autogenous shrinkage could be clearly mitigated. In addition, the solid alkaline activator provided a convenient way of preparation and curing and reduced damage from alkaline corrosion;The addition of NaAlO_2_ slowed the early hydration reaction, while made the later hydration reaction more complete, reducing the total porosity, changing the pore-size distribution and having a significant impact on the shrinkage performance;Added NaAlO_2_ could generate hydrotalcite and tetranatrolite, and the generation increased with increased added NaAlO_2._ Moreover, the change of the generated phase composition clearly mitigated shrinkage of AAS;For AAS with different Na_2_O contents, the evaporation from the paste played a major role in drying shrinkage and the degree of hydration reaction mainly contributed to autogenous shrinkage.

## Figures and Tables

**Figure 1 materials-13-03499-f001:**
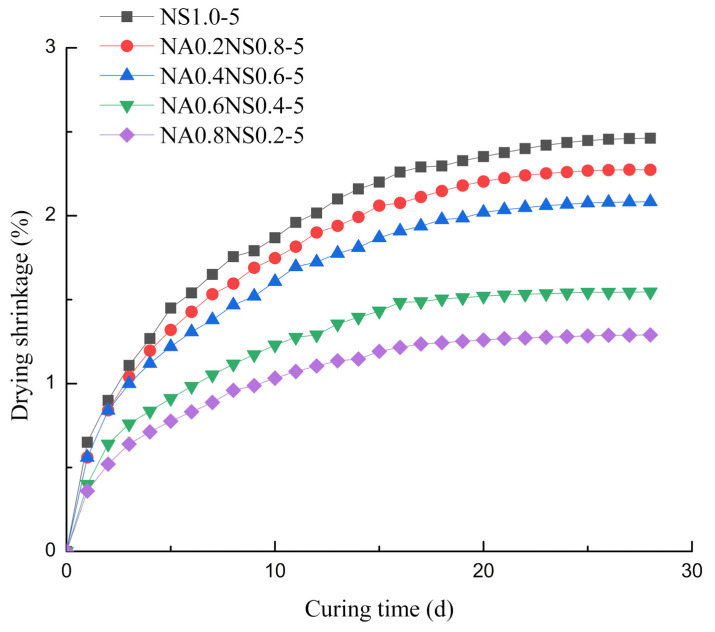
Effect of different NaAlO_2_ substitution rates on drying shrinkage.

**Figure 2 materials-13-03499-f002:**
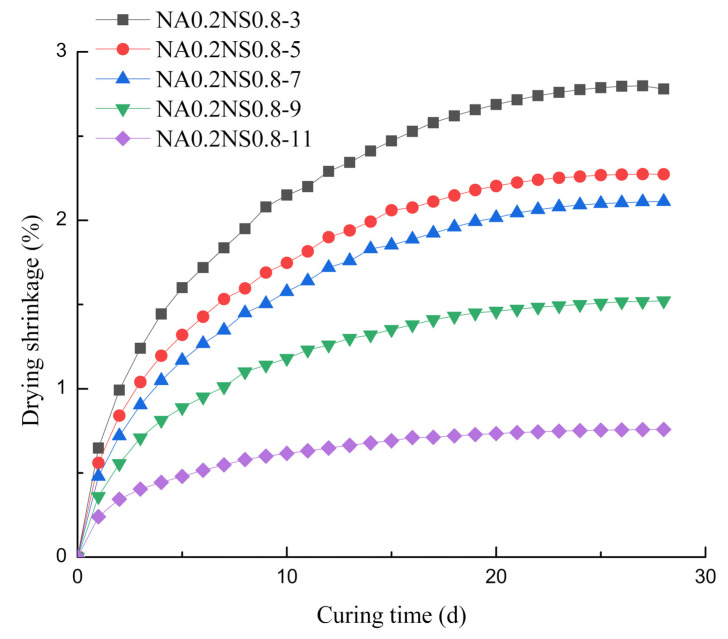
Effect of different Na_2_O contents on drying shrinkage.

**Figure 3 materials-13-03499-f003:**
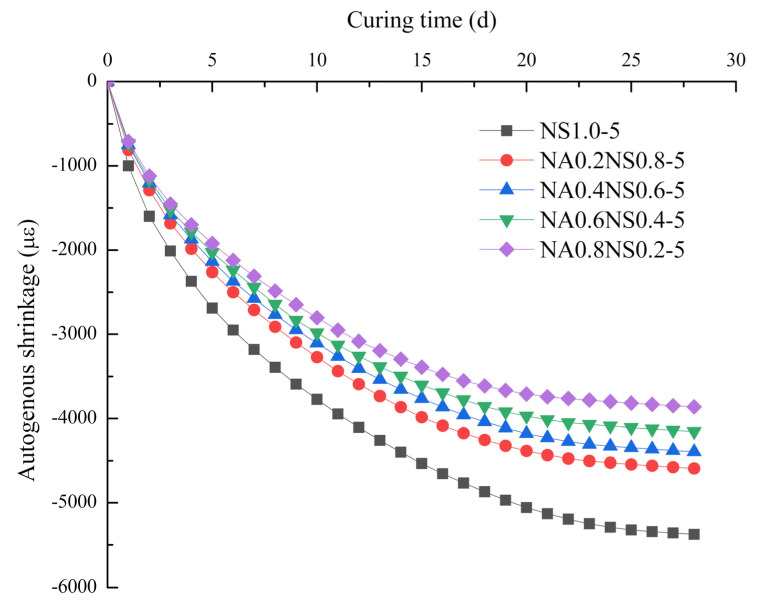
Effect of different NaAlO_2_ substitution rates on autogenous shrinkage.

**Figure 4 materials-13-03499-f004:**
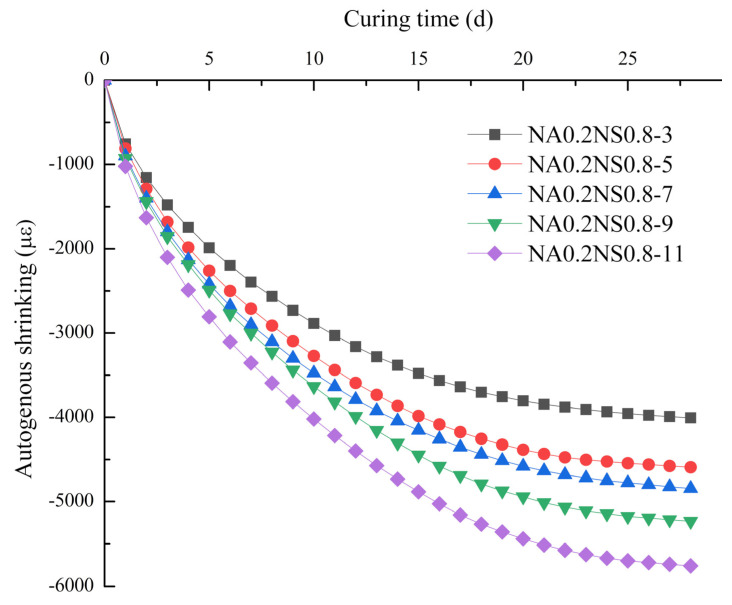
Effect of different Na_2_O contents on autogenous shrinkage.

**Figure 5 materials-13-03499-f005:**
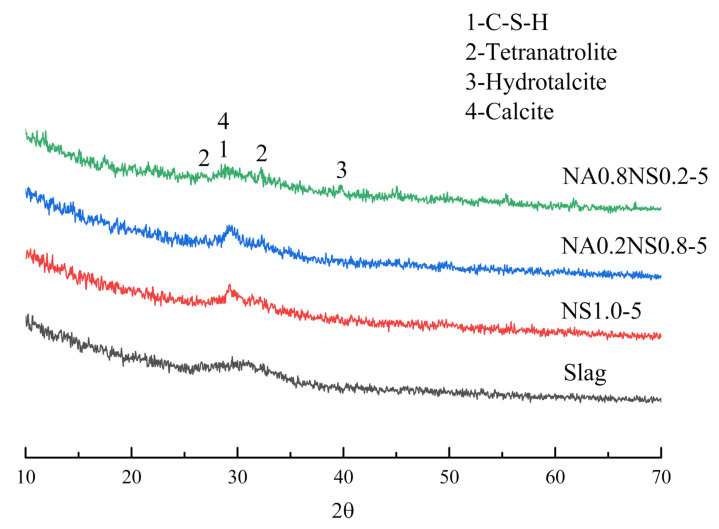
XRD image of raw slag and different NaAlO_2_ substitution rates at 7 d.

**Figure 6 materials-13-03499-f006:**
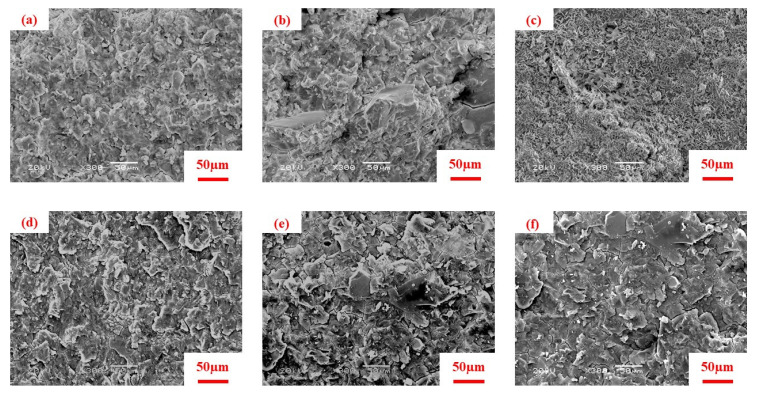
Microscopic structure of AAS with different NaAlO_2_ substitution rates at 7 d and 28 d. (**a**) NS1.0-5 at 7 d; (**b**) NA0.2NS0.8-5 at 7 d; (**c**) NA0.8NS0.2-5 at 7 d; (**d**) NS1.0-5 at 28 d; (**e**) NA0.2-NS0.8-5 at 28 d; (**f**) NA0.8NS0.2-5 at 28 d.

**Figure 7 materials-13-03499-f007:**
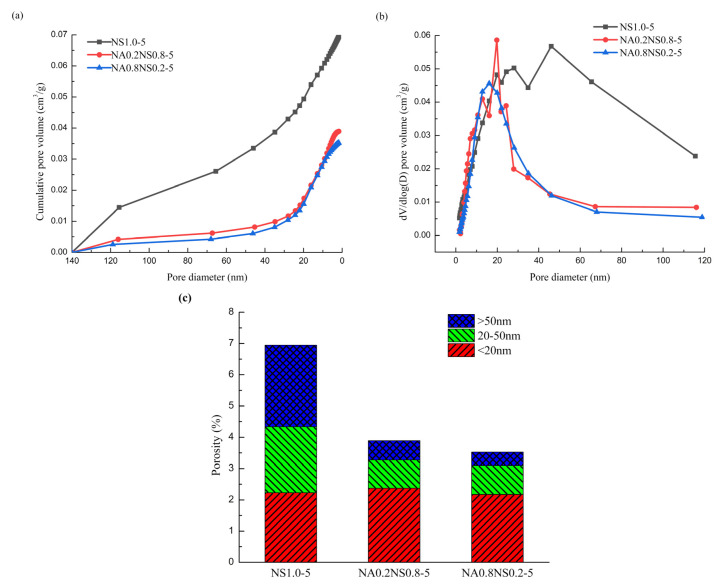
Pore-size distribution of AAS with different NaAlO_2_ substitution rates. (**a**) Cumulative intrusion; (**b**) different intrusion; (**c**) porosity.

**Figure 8 materials-13-03499-f008:**
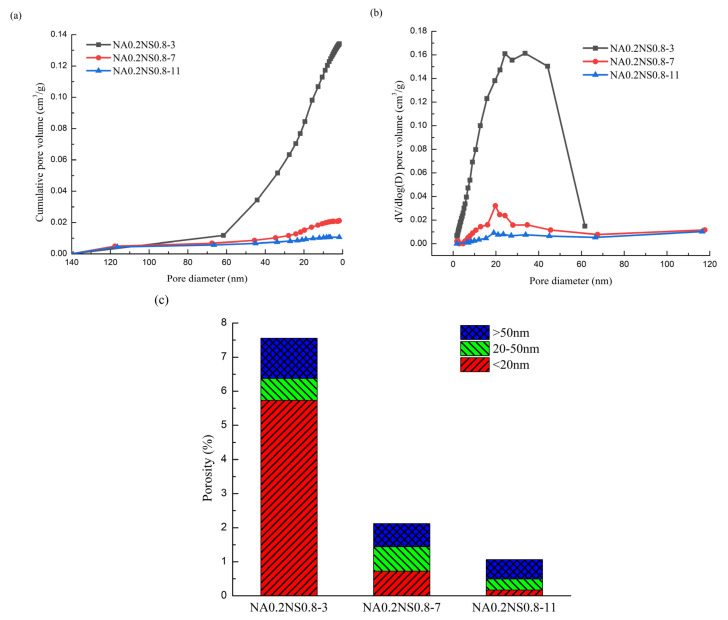
Pore-size distribution of AAS with different Na_2_O contents. (**a**) Cumulative intrusion; (**b**) different intrusion; (**c**) porosity.

**Table 1 materials-13-03499-t001:** Chemical components of blast furnace slag (%).

Oxide	CaO	SiO_2_	MgO	Al_2_O_3_	MnO	TiO_2_	S	FeO
Percentage/%	40.91	32.82	7.87	14.93	0.48	0.92	0.91	1.16

**Table 2 materials-13-03499-t002:** Mix proportions the prepared alkali-activated slag (AAS).

Code	NaAlO_2_ Substitution Rate (%)	Na_2_O Content (%)	Slag (g)	NaAlO_2_ (g)	Na_2_SiO_3_ (g)	H_2_O (g)
NS1.0-5	0	5	2000	0.0	458.7	700
NA0.2NS0.8-5	20	5	2000	52.9	366.9	700
NA0.4NS0.6-5	40	5	2000	105.9	275.2	700
NA0.6NS0.4-5	60	5	2000	158.7	183.5	700
NA0.8NS0.2-5	80	5	2000	211.6	91.7	700
NA0.2NS0.8-3	20	3	2000	31.7	220.1	700
NA0.2NS0.8-7	20	7	2000	74.1	513.7	700
NA0.2NS0.8-9	20	9	2000	95.2	660.5	700
NA0.2NS0.8-11	20	11	2000	116.4	807.3	700

**Table 3 materials-13-03499-t003:** Volume proportions of pore structure distribution of AAS.

Code	Total Porosity (%)	Volume Proportions (%)		
		Micropores	Mesopores	Macropores
NS1.0-5	6.94	32.06	30.36	37.58
NA0.2 NS0.8-5	3.90	60.89	23.20	15.91
NA0.8 NS0.2-5	3.53	61.67	26.27	12.06
NA0.2 NS0.8-3	13.41	75.92	8.45	15.63
NA0.2 NS0.8-7	2.12	34.39	33.63	31.98
NA0.2 NS0.8-11	1.07	16.24	30.14	53.62

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
