# Peer review of "Mitigating the Drying Shrinkage and Autogenous Shrinkage of Alkali-Activated Slag by NaAlO2"

_materials, 2020, doi:10.3390/ma13163499_

Round 1
Reviewer 1 Report
This paper (Mitigating the drying shrinkage and autogenous shrinkage of alkali-activated slag by NaAlO2) presents interesting results but needs a thorough revision before being considered for publication. Some sections need to be completely rewritten, like the Introduction, literature review and Discussion.
Introduction: The theoretical, analytical and standard approaches should be discussed.
The novelties have to be outlined. It has to be completely rewritten so that the focus of the work and its innovative content can be really appreciated.
Literature review: The Literature review is now a mere list of information but the authors have to provide their own "unifrying" view and not only citing previous work.
In addition, in a quick search I found a number of papers on this topic that you did not cite. I am listing them here, please consider them in the literature review and in the interpretation of the results:
- Doubts over capillary pressure theory in context with drying and autogenous shrinkage of alkali-activated materials, Construction and Building Materials, Volume 248, 10 July 2020, Article 118620, Lukáš Kalina, Vlastimil Bílek, Eva Bartoníčková, Michal Kalina, Radoslav Novotný
- The combined use of admixtures for shrinkage reduction in one-part alkali activated slag-based mortars and pastes, Construction and Building Materials, Volume 248, 10 July 2020, Article 118682, L. Coppola, D. Coffetti, E. Crotti, S. Candamano, T. Pastore
- Effect of internal curing by superabsorbent polymers – Internal relative humidity and autogenous shrinkage of alkali-activated slag mortars, Construction and Building Materials, Volume 123, 1 October 2016, Pages 198-206, Chiwon Song, Young Cheol Choi, Seongcheol Choi
- Shrinkage and strength development of alkali-activated fly ash-slag binary cements, Construction and Building Materials, Volume 150, 30 September 2017, Pages 808-816, Maryam Hojati, Aleksandra Radlińska
- Shrinkage mitigation strategies in alkali-activated slag, Cement and Concrete Research, Volume 101, November 2017, Pages 131-143, Hailong Ye, Aleksandra Radlińska
Results and discussion: The paper presents a few amount of results from usual experiments but without a theoretical and practical approach.
Conclusions: The discussion about technological benefit have to be separated in the article according points of conclusions. The analysis of the results is quite basic and deserves better and deeper processing.
Reviewer 2 Report
The paper examines with scientific method the effect of the composition of slag mixtures on the microporosity of solid structures. It would be interesting to highlight the consequences in terms of strenght, even after the 28 days examine.
Sometimes, the differences between the measured shrinkages are slight; the introduction of statistical comparisons would be helpful.
Reviewer 3 Report
Thanks to the authors for performing this research. The following comments need to be addressed:
- In line 15, check the grammar for “which had significantly effects”.
- In line 18 and 19, is “Moreover, the effectiveness of Na2O content on drying shrinkage and autogenous shrinkage.” a complete sentence?
- In line 20, is “leaded” a correct verb?
- Check the grammar for the last sentence of the Abstract as well.
- The Introduction is also full of unclear sentences with grammar issues.
- The rest of the paper contains numerous language problems as well.
In summary, the manuscript is not acceptable in the current form for publication. It needs to be professionally proofread before resubmitting for review.
Reviewer 4 Report
Dear Authors,
This paper presents an experimental study on the effect of using the effect of NaAlO2 in shrinkage of the alkali-activated slag (AAS). Specifically, the study investigated the effect of NaAlO2 in mitigating drying shrinkage and the autogenous shrinkage of AAS. SEM, XRD characterized the structure and properties of AAMTs. The effect of using different amounts of NaAlO2 investigated by measuring the drying shrinkage and porosity measurement. Although the paper addresses an important topic related to shrinkage of alkali-activated slag and should be interesting to the authors of Materials, it has various limitations as commented below:
- Some grammatical errors make the paper difficult to understand. Please revise the paper carefully to correct them.
- In the introduction, you need to clearly explain the final application of this research and say why this research wanted to understand the mechanism of hydration of mitigated shrinkage.
- In the materials and method, you need to describe the synthesis of the specimens. In the last part of the materials and method, you need to have a sentence for clarification of your sample codes. The sentence like this: The samples are named as NAXNSX-Y [where x relates to the…..].
- The SEM images are not clear; it should be space or border between each figure to detect them.
- The SEM images always presented with a clear label and scale bar; your images did not have any of these requirements. The scale bar should be inserted into the figure. I would recommend you insert all 6 figures in one part to compare the 7 days and 28 days of curing.
- The BET part explained well, but the figure 8 and 9 are similar, please check the graphs, these mistakes are not acceptable in the research paper.
Round 2
Reviewer 1 Report
The article can be publish in actual version.
Reviewer 3 Report
Thanks to the authors for submitting the revision. The technical revisions enhanced the merit of manuscript and some changes have been made to the English writing. I suggest the authors to carefully review the English prior to the final publication as there are still many language issues. Low quality writing may affect the interest of the paper to the readers.
Reviewer 4 Report
Dear Authors,
Thank you for approving the changes and revising the manuscript.
Please get the manuscript language check as it has still some typo and grammatical errors.
Bests,
Reviwer
